# Exploring the Interplay of Antioxidants, Inflammation, and Oxidative Stress: Mechanisms, Therapeutic Potential, and Clinical Implications

**DOI:** 10.3390/diseases13090309

**Published:** 2025-09-22

**Authors:** Sumayyah Yousef Altanam, Nedal Darwish, Ahmed Bakillah

**Affiliations:** 1Department of Chemistry, College of Science, King Faisal University (KFU), Al Ahsa 36362, Saudi Arabia; loaloah.78@gmail.com; 2Department of Hematology and Oncology, Rochester General Hospital, Rochester, NY 14621, USA; nedal.darwish@rochesterregional.org; 3Biomedical Research Core-A, King Abdullah International Medical Research Center (KAIMRC), King Saud bin Abdulaziz University for Health Sciences (KSAU-HS), Al Ahsa 36428, Saudi Arabia; 4King Abdulaziz Hospital, Ministry of National Guard-Health Affairs (MNG-HA), Al Ahsa 36428, Saudi Arabia

**Keywords:** oxidative stress, reactive oxygen species, inflammation, antioxidants, anti-inflammatory agents, diseases, nutraceuticals, NRF2–Keap1 pathway, NF-κB

## Abstract

Oxidative stress, resulting from an imbalance between reactive oxygen species (ROS) production and antioxidant defenses, is a major factor in chronic diseases such as cardiovascular disorders, neurodegeneration, diabetes, and cancer. Despite extensive research, current reviews often discuss antioxidants or inflammatory pathways separately, which limits their translational impact. The primary objective of this review is to present an integrated analysis of oxidative stress and inflammation, connecting molecular mechanisms with clinical evidence. We focus on the dual roles of natural and synthetic antioxidants in managing redox balance, regulating inflammatory signaling, and targeting new molecular pathways. Unlike previous work, this review emphasizes recent clinical findings, ongoing therapeutic challenges, and innovative strategies, including combination approaches and synthetic derivatives designed to improve effectiveness. By combining biochemical, preclinical, and clinical perspectives, we highlight both established knowledge and critical gaps. Ultimately, this review highlights the clinical significance of redox biology, clarifies the potential of antioxidant-based treatments, and outlines future research directions essential for translating these insights into effective therapies for chronic disease management.

## 1. Introduction

In recent decades, the scientific community has paid increasing attention to the complex relationship between oxidative stress, inflammation, and the development of chronic diseases [1,2,3]. Oxidative stress and chronic inflammation have been implicated in a wide range of pathological conditions, including cardiovascular diseases, kidney disease, diabetes, neurodegenerative diseases, and even cancer. As we face an era of rapid environmental changes, unhealthy lifestyles, and population aging, understanding the molecular mechanisms underlying oxidative stress and inflammation is more critical than ever. In this context, the exploration of potential therapeutic agents, particularly nutraceuticals, has gained significant traction. These bioactive compounds, often derived from natural sources, hold promise in modulating oxidative stress and inflammation to prevent or manage various diseases [4,5,6,7].

The main goal of this review is to explore the connections between oxidative stress and inflammation, with an emphasis on how nutraceuticals can influence these processes. We will review recent research on various antioxidant compounds including polyphenols, omega-3 fatty acids, and other bioactive molecules, and their impact on oxidative stress and inflammation. Furthermore, this review will discuss the challenges and opportunities of integrating nutraceuticals into clinical practice, with a focus on their bioavailability, optimal dosages, and potential for personalized treatments.

This review aims to provide a comprehensive understanding of the molecular mechanisms linking oxidative stress and inflammation, while highlighting promising nutraceutical interventions that could lead to new strategies for preventing and managing chronic diseases.

The interplay between oxidative stress, antioxidants, and inflammation forms a dynamic network that influences the progression of disease. On the one hand, ROS are critical for immune signaling and play a significant role in activating inflammatory pathways. However, when ROS production exceeds the body’s ability to neutralize them, it leads to tissue damage, amplifying the inflammatory response and fostering disease development. In this context, antioxidants play a pivotal role in breaking this harmful cycle. By scavenging ROS, antioxidants help mitigate oxidative damage and reduce inflammation, offering a protective mechanism [5,8,9].

Additionally, certain antioxidants have been shown to influence inflammation directly. For example, polyphenolic compounds such as curcumin and resveratrol have been shown to inhibit the activation of key inflammatory pathways, including NF-κB and MAPK, which are involved in cytokine production [10,11]. By modulating both oxidative stress and inflammation, antioxidants may offer a dual strategy to prevent or alleviate chronic diseases, providing a promising approach for therapeutic interventions. Understanding these mechanisms is essential for developing more effective strategies to manage the effects of oxidative stress and inflammation in chronic diseases.

Although traditional pharmacological treatments are common, they often lead to side effects and may not target the underlying causes of these diseases. Consequently, interest in natural compounds with antioxidants and anti-inflammatory properties has grown recently as potential therapeutic options [12,13].

## 2. Oxidative Stress: Mechanisms and Pathophysiology

### 2.1. Endogenous Sources

The formation of intracellular free radicals in mitochondria and peroxides occurs due to basic metabolism during normal oxidative phosphorylation, which is essential for ATP generation (Table 1). Cytosolic enzyme systems also produce these radicals. In addition, ROS are enzymatically generated in macrophages to destroy pathogens. However, their excessive generation weakens host cells and organs due to the inability to distinguish between host cells and infectious agents [14].

Highly reactive molecules of ROS and reactive nitrogen species (RNS) promote random damage to cellular structures, leading to cell death by interrupting the physiological functions of key biomolecules [8]. The most common biologically active radicals and molecules that may induce pathophysiological conditions include non-radical hydrogen peroxide (H_2_O_2_), superoxide anion radical (O_2_•^−^), hydroxyl radical (OH•), nitric oxide radical (NO•), and peroxynitrite (ONOO^−^).

ATP is produced in mitochondria through energy conversion in a usable form by cells. In this process, oxidative phosphorylation involves the electron transport chain in which hydrogen ions are moved across the inner mitochondrial membrane. Electrons pass through a series of redox-active proteins, each having a higher reduction potential than the last, eventually reducing oxygen to water. However, approximately 1–3% of electrons leak, leading to the premature and partial reduction of oxygen and the formation of O_2_•^−^ [6,15].

Enzymatic and non-enzymatic reactions produce superoxide radicals and are highly reactive. Their protonated form, HO_2_•, easily penetrates the phospholipid bilayer, initiating lipid oxidation. This radical can also reduce iron complexes such as Cytochrome c and ferric-EDTA, acting as a potent reducing agent. Conversely, it can oxidize substances such as ascorbic acid and tocopherol which participate in numerous reactions that generate more radical and oxidative species.

Hydroxyl radicals are generated by the reaction of O_2_•^−^ with H_2_O_2_. OH• is regarded as the most harmful ROS, as it reacts aggressively with lipids, proteins, and DNA, causing significant cell damage. It can abstract an electron from polyunsaturated fatty acids, triggering lipid peroxidation [16].

In vivo, H_2_O_2_ is formed via a dismutation reaction catalyzed by the enzyme superoxide dismutase (SOD), or by metabolic oxygen consumption in peroxisomes. At low concentrations, H_2_O_2_ damages cells, while high concentrations inactivate energy-producing enzymes such as glyceraldehyde-3-phosphate dehydrogenase. It can also affect biological membranes by easily penetrating them [17].

ONOO^−^ is a highly toxic compound that results from the reaction between O_2_•^−^ and NO•. ONOO^−^ readily reacts with CO_2_, O_2_, and water. Its reaction with CO_2_ produces highly reactive intermediates such as nitroso-peroxy-carboxylate (ONOOCO_2_^−^) and peroxynitrous acid (ONOOH), which decompose to form hydroxyl (OH•) and nitrogen dioxide (NO_2_•) radicals. ONOO^−^ can damage cells by oxidizing lipids, methionine, and tyrosine residues in proteins, and DNA, forming nitroguanine [18].

**Table 1 diseases-13-00309-t001:** Major reactive species, their primary cellular sources, and mechanisms.

Reactive Species/Antioxidants	Primary Production Source (Subcellular Location/Enzyme System)	Reaction/Mechanism	References
Superoxide (O_2_•^−^)	Mitochondrial electron transport chain (Complexes I & III), NADPH oxidases (NOX family), xanthine oxidase, and uncoupled NOS.	One-electron reduction of O_2_ → O_2_^−^; rapidly dismutates by SOD to H_2_O_2_.	[19,20]
Hydrogen peroxide (H_2_O_2_)	Product of SOD-mediated dismutation (cytosol/mitochondria), peroxisomal oxidases, and some NOX activity.	Two-electron product (dismutation or direct 2-e^−^ reduction of O_2_); diffusible signaling oxidant; detoxified by catalase and glutathione peroxidases.	[21,22]
Hydroxyl radical (•OH)	Generated locally from H_2_O_2_ via iron-mediated Fenton/Haber–Weiss chemistry (labile Fe^2+^ pools).	H_2_O_2_ + Fe^2+^ → •OH + OH^−^ (Fenton); extremely reactive and non-selective.	[23,24]
Nitric oxide (NO)	Nitric oxide synthases (eNOS, iNOS, nNOS)—cytosolic/membrane-associated.	Radical gasotransmitter; reacts rapidly with O_2_•^−^ to form peroxynitrite (ONOO^−^).	[25]
Peroxynitrite (ONOO^−^/ONOOH)	Formed by diffusion-limited reaction between NO• and O_2_•^−^ in cytosol/near membranes.	Potent oxidant/nitrating species; yields secondary radicals (NO_2_, CO_3_•^−^) and modifies proteins/lipids.	[26,27]
Lipid peroxyl radical (LOO)/lipid hydroperoxides (LOOH)	Initiated when ROS attack polyunsaturated fatty acids in membranes or lipoproteins (e.g., LDL)—membrane/LDL surface.	Radical chain-propagation (L• → + O_2_ → LOO → abstracts H → LOOH); leads to reactive aldehydes (MDA, 4-HNE) and ox-LDL formation.	[28,29]
Antioxidant enzymes (SOD, Catalase, GPX)	SOD1 (cytosol), SOD2 (mitochondria), catalase (peroxisomes), GPXs (cytosol/mitochondria).	SOD: 2 O_2_•^−^ + 2 H^+^ → H_2_O_2_ + O_2_. Catalase/GPX: H_2_O_2_ → H_2_O (via 2 e^−^ reduction or catalase decomposition).	[30,31]

### 2.2. Exogenous Sources

Environmental and external factors play a significant role in causing oxidative stress by increasing reactive oxygen and nitrogen species in the body (Figure 1). For instance, cigarette smoke introduces both superoxide (O_2_•^−^) and nitric oxide (NO•), directly elevating free radical levels. Similarly, ozone exposure leads to lipid oxidation and promotes neutrophil infiltration into the airway lining. Even low levels of ozone exposure can trigger the release of inflammatory and cytotoxic mediators, including myeloperoxidase (MPO), eosinophil cationic protein, lactate dehydrogenase, and albumin, all of which collectively raise ROS in the lungs and other tissues due to the oxygen-rich environment. Heavy metals such as iron, copper, cadmium, mercury, nickel, lead, and arsenic further exacerbate oxidative stress by catalyzing lipid peroxidation and interacting with proteins and DNA, impairing enzyme activity and damaging cells. Ionizing radiation intensifies these effects through multiple pathways: when oxygen is present, it produces OH, O_2_•^−^, and organic radicals, which can then form H_2_O_2_ and organic hydroperoxides via Fenton reactions. Moreover, radiation interacts with water, making up ~ 55–60% of the human body, through radiolysis, generating OH, H_2_O_2_, O_2_•^−^, and O_2_, all of which contribute to oxidative stress at both cellular and tissue levels. These external sources of oxidative stress highlight the importance of the body’s internal antioxidant defense systems [32], which encompass both enzymatic and non-enzymatic mechanisms by maintaining redox balance and preventing cellular damage.

### 2.3. Impact of Oxidative Stress on Cellular Structures and Functions

The harmful effects of oxidative stress on cellular structures are significant. ROS and RNS can directly damage lipids, proteins, and nucleic acids, resulting in structural and functional abnormalities. Lipid peroxidation, a process in which ROS attack lipid membranes, produces toxic aldehydes that can compromise cell membrane integrity and fluidity. This disruption can hinder cell signaling and lead to cellular dysfunction [33].

Protein oxidation occurs when ROS modifies amino acid side chains, resulting in changes to protein structure and function. This can lead to the loss of enzyme activity, disrupted protein–protein interactions, and the buildup of misfolded proteins, which are often linked to neurodegenerative diseases such as Alzheimer’s and Parkinson’s [34]. DNA damage caused by ROS can cause mutations, chromosomal fragmentation, and even apoptosis. If left unrepaired, these mutations may contribute to the onset and advancement of cancer [35].

Furthermore, oxidative stress has a significant impact on cellular signaling pathways. For example, ROS activates transcription factors such as nuclear factor-kappa B (NF-κB) and activator protein-1 (AP-1), which regulate the expression of genes involved in inflammation, cell survival, and apoptosis [36]. This activation may lead to chronic inflammation, potentially exacerbating tissue damage and disease progression.

Oxidative stress is also a key factor in aging and age-related diseases. As the body’s antioxidant defenses weaken with age, the accumulation of oxidative damage to cellular components plays a significant role in the aging process and the development of age-associated conditions, such as atherosclerosis, cataracts, and neurodegeneration. In particular, mitochondria, which are the primary source of ROS, undergo damage over time, resulting in impaired energy production and increased ROS generation, which creates a vicious cycle of mitochondrial dysfunction and cellular aging [37]. The mechanisms behind oxidative stress are complex, involving multiple sources of ROS and RNS, as well as advanced antioxidant defenses. When these systems fail to maintain redox balance, oxidative stress can cause significant cellular damage, contributing to the development of various chronic diseases [38]. Understanding these processes, along with the role of antioxidants in mitigating oxidative damage, offers promising therapeutic options for preventing and treating diseases associated with oxidative stress.

Future research should continue exploring new antioxidant-based treatments and strategies to enhance the body’s natural defenses, ultimately leading to better health outcomes despite oxidative stress and its related conditions.

### 2.4. Endogenous Antioxidant Defense Systems (Enzymatic and Non-Enzymatic)

The endogenous antioxidant defense system consists of a carefully regulated balance between enzymatic and non-enzymatic molecules, acting as the first line of defense against redox imbalance and oxidative damage (Table 2). Enzymatic antioxidants, such as superoxide dismutase (SOD), catalase (CAT), and glutathione peroxidases (GPx), work together to neutralize reactive oxyROS and RNS [39]. SOD converts O_2_^−^• into H_2_O_2_, which, if not quickly neutralized, can participate in Fenton reactions, producing highly reactive hydroxyl radicals. Then, CAT, mainly found in peroxisomes, and GPx, primarily active in mitochondria and the cytosol, convert H_2_O_2_ into harmless water and oxygen, preventing oxidative damage to nucleic acids, proteins, and membrane lipids.

Importantly, each enzyme requires specific cofactors to work. SOD types need manganese, copper, or zinc, while CAT contains a heme iron center, and GPx activity depends on selenium. Changes in the levels or activity of these enzymes are associated with various health issues, including neurodegeneration, cancer progression, and heart problems. Recent studies indicate that increasing SOD activity or administering CAT through gene therapy can reduce neuronal damage after ischemia and lower infarct size [40,41]. Additionally, synthetic mimetics of SOD and mitochondrial-targeted antioxidants such as mitoquinone (MitoQ) or plastoquinone (SkQ1) are being tested in clinical settings to regulate redox balance in chronic diseases.

Non-enzymatic antioxidants, including GSH, α-lipoic acid, melatonin, uric acid, bilirubin, and coenzyme Q10, work by directly scavenging free radicals or restoring oxidized antioxidants. GSH, a tripeptide made up of glutamate, cysteine, and glycine, functions as a key redox buffer and serves as a substrate for GPx. In conditions like Parkinson’s and Alzheimer’s diseases, GSH depletion is linked to increased ROS levels and neuronal cell death. Supplementation with N-acetylcysteine (NAC) or liposomal GSH has shown promising results in restoring cellular redox balance [42]. Similarly, α-lipoic acid, through its disulfide/dithiol redox pair, not only neutralizes SOD, OH•, and ROO• radicals but also regenerates other antioxidants like GSH and vitamin C, making it a valuable addition in managing diabetic neuropathy and metabolic syndrome [43].

Melatonin, mainly produced by the pineal gland, has notable antioxidant effects that go beyond regulating the sleep–wake cycle. Its amphiphilic nature enables it to cross cell membranes easily and gather within mitochondria, where it combats oxidative stress and prevents the opening of the mitochondrial permeability transition pore (mPTP). Clinical studies have shown its potential to improve outcomes in cases of sepsis, ischemic injuries, and even post-COVID-19 fatigue [44].

Although substantial experimental evidence supports the effectiveness of endogenous antioxidant systems, their clinical use is limited by pharmacokinetic challenges, including low oral bioavailability, poor tissue targeting, and a short half-life. Additionally, systemic antioxidant administration can disrupt normal redox signaling, emphasizing the need for interventions tailored to specific contexts. Advances include nano-formulated antioxidants, prodrug strategies, and redox phenotype-guided therapies. Personalized antioxidant approaches that consider disease stage, genetic background, and redox status could improve treatment outcomes.

### 2.5. Role of Antioxidants in Organ Protection

Oxidative stress is a common pathogenic mechanism that connects metabolic disorders, cardiovascular disease, chronic kidney disease, gastrointestinal injury, and neurodegeneration. Antioxidants, both natural and synthetic, exert protective effects by neutralizing ROS, maintaining redox balance, and regulating inflammatory and apoptotic pathways.

The cardiovascular system is particularly vulnerable to oxidative stress due to continuous exposure to oxidized lipoproteins and vascular inflammation. Antioxidants such as vitamin E, vitamin C, polyphenols (resveratrol, curcumin, green tea catechins), and pharmacological agents with antioxidant activity (e.g., statins, probucol) reduce LDL oxidation, improve endothelial nitric oxide bioavailability, and attenuate atherogenesis [45,46,47]. Clinical studies have shown that diets rich in natural antioxidants correlate with a lower incidence of coronary artery disease and improved vascular function [48].

The kidney is also vulnerable to ROS-mediated damage in both acute and chronic injury. Antioxidants, including N-acetylcysteine, curcumin, and green tea polyphenols, reduce ischemia–reperfusion injury, attenuate tubular apoptosis, and decrease proteinuria in experimental nephropathy [49,50]. Synthetic antioxidants have been used to protect against contrast-induced nephropathy and drug-induced renal toxicity. Thus, targeting oxidative stress pathways is increasingly recognized as a potential renoprotective strategy.

The gastrointestinal mucosa is continuously exposed to dietary oxidants, microbial metabolites, and inflammatory mediators. Flavonoids, carotenoids, and vitamins (C and E) strengthen mucosal defenses by enhancing epithelial barrier integrity, scavenging ROS, and suppressing NF-κB activation [51,52]. Experimental studies demonstrate that curcumin and quercetin reduce the severity of colitis, while vitamin C and other antioxidants accelerate gastric ulcer healing [53,54,55].

The brain consumes high amounts of oxygen and is rich in polyunsaturated fatty acids, which makes it very vulnerable to lipid peroxidation and oxidative injury. Natural antioxidants such as polyphenols, omega-3 fatty acids, and coenzyme Q10, as well as synthetic antioxidants, have shown neuroprotective effects by preventing protein aggregation, reducing mitochondrial dysfunction, and suppressing neuroinflammation [56,57]. Furthermore, clinical and preclinical studies support their role in slowing cognitive decline in Alzheimer’s disease and reducing dopaminergic neuronal loss in Parkinson’s disease [58,59].

In summary, antioxidants play a crucial role in protecting vital organs from oxidative stress–induced injury. Their multifaceted actions on redox signaling, inflammation, and apoptosis highlight their potential as supportive therapeutics in preventing and managing metabolic and degenerative diseases.

## 3. Inflammation and Its Crosstalk with Oxidative Stress

Inflammation and oxidative stress are closely connected biological processes that play a central role in the development of many chronic diseases. Oxidative stress, characterized by an imbalance between the production of ROS and antioxidant defenses, can trigger various inflammatory pathways, including the nuclear factor kappa B (NF-κB) signaling pathway. NF-κB is a key transcription factor that regulates the expression of pro-inflammatory cytokines such as tumor necrosis factor-alpha (TNF-α), interleukin-6 (IL-6), and interleukin-1β (IL-1β), thereby amplifying the inflammatory response [37,60]. Conversely, inflammation can further increase ROS production by activating immune cells, such as macrophages and neutrophils, which release oxidative molecules as part of the host’s defense mechanisms. This bidirectional interaction creates a vicious cycle that perpetuates cellular damage, tissue dysfunction, and contributes to the development and progression of chronic conditions such as cardiovascular diseases, diabetes mellitus, kidney diseases, neurodegenerative disorders, and cancer [61,62]. Mounting evidence indicates that prolonged activation of inflammatory and oxidative pathways disrupts cellular balance, worsens mitochondrial dysfunction, and ultimately perpetuates disease. Therefore, targeting the molecular interaction between oxidative stress and inflammation presents a promising therapeutic strategy to reduce the impact of chronic inflammatory diseases.

## 4. The NRF2–Keap1 Pathway and Its Cross-Talk with Inflammatory Signaling

The nuclear factor erythroid 2–related factor 2 (NRF2) is a key transcription factor that controls the expression of a wide range of antioxidant and cytoprotective genes. Under normal conditions, NRF2 is kept in the cytoplasm by Kelch-like ECH-associated protein 1 (Keap1), which marks it for ubiquitination and subsequent degradation by the proteasome. When cells face oxidative stress or electrophilic compounds, critical cysteine residues in Keap1 become modified, causing conformational changes that reduce NRF2 ubiquitination. This stabilization allows NRF2 to accumulate and move into the nucleus, where it binds to antioxidant response elements (ARE) in the promoter regions of target genes [63]. This leads to increased transcription of phase II detoxifying enzymes and antioxidant proteins, such as heme oxygenase-1 (HO-1), NAD(P)H: quinone oxidoreductase 1 (NQO1), the catalytic (GCLC) and modifier (GCLM) subunits of glutamate-cysteine ligase, and glutathione peroxidase (GPx). Collectively, these effectors boost redox balance, protect against lipid peroxidation, and reduce oxidative damage [64].

Importantly, the NRF2–Keap1 pathway does not operate alone but interacts with pro-inflammatory signaling pathways, especially the NF-κB pathway. Oxidative stress strongly activates NF-κB, whereas NRF2 activation counteracts this by lowering ROS levels and inhibiting NF-κB–dependent gene expression (Figure 2). Additionally, direct molecular interactions between these pathways have been identified. For example, the NF-κB p65 subunit can suppress NRF2’s transcriptional activity by competing for coactivators, while NRF2-driven HO-1 expression can limit NF-κB activation through carbon monoxide and biliverdin signaling [65,66]. This reciprocal regulation maintains a delicate balance between antioxidant defenses and inflammatory responses (Figure 2).

Dysregulated NRF2–Keap1 signaling has been linked to the development of chronic diseases marked by oxidative stress and low-grade inflammation, including cardiovascular disease, type 2 diabetes mellitus, neurodegeneration, and cancer. Therefore, therapeutic approaches aimed at activating NRF2, such as electrophilic compounds, natural antioxidants like curcumin and sulforaphane, and synthetic derivatives, are actively being explored for their potential to restore redox balance and reduce inflammatory injury [64,67,68].

Under resting conditions, nuclear factor erythroid 2–related factor 2 (NRF2) is held in the cytoplasm by Kelch-like ECH-associated protein 1 (Keap1). Oxidative or conformational changes in Keap1 prevent NRF2 from being ubiquitinated, allowing NRF2 to stabilize and accumulate in the nucleus. Once inside, NRF2 forms a heterodimer with small Maf proteins and binds to antioxidant response element (ARE) sequences to activate a wide range of cytoprotective genes, including heme oxygenase-1 (HO-1), NAD(P)H quinone oxidoreductase 1 (NQO1), glutathione S-transferases (GSTs), superoxide dismutase (SOD), catalase (CAT), glutathione peroxidase (GPx), and enzymes involved in glutathione synthesis. These gene products work together to neutralize reactive oxygen species (ROS), repair oxidative damage, and facilitate detoxification, thus maintaining redox and metabolic balance. Simultaneously, inflammatory signals activate nuclear factor kappa B (NF-κB), promoting the transcription of pro-inflammatory cytokines and mediators. The NRF2 and NF-κB pathways interact through bidirectional cross-talk. Activation of NRF2 reduces NF-κB–driven inflammation, whereas chronic inflammatory signals can impair NRF2 activity through post-translational and transcriptional mechanisms. Disruption of this balance due to abnormal Keap1 activity, genetic mutations, persistent oxidative or inflammatory stress, or mitochondrial dysfunction weakens ARE-dependent defenses, leading to excessive ROS buildup, lipid peroxidation, protein carbonylation, DNA strand breaks, and compromised detoxification capacity. These processes drive the development and progression of chronic diseases, such as cardiovascular disease, type 2 diabetes mellitus, neurodegenerative disorders, chronic kidney disease, and cancer. This illustration was created using Canva (https://www.canva.com, accessed on 12 September 2025).

## 5. Types and Mechanisms of Antioxidants

Nutraceutical and pharmacological antioxidants are crucial in combating oxidative stress, a condition associated with the development of numerous chronic diseases. Nutraceutical antioxidants, derived from natural dietary sources such as flavonoids, vitamins, and carotenoids, help neutralize free radicals and support the body’s natural defense mechanisms. Conversely, pharmacological antioxidants are synthetically created to enhance their antioxidant effects through improved stability, increased bioavailability, and enhanced cellular targeting. These agents have demonstrated promising results in managing oxidative damage related to aging, neurodegenerative diseases, and cardiovascular conditions [69,70].

### 5.1. Nutraceuticals

Nutraceuticals are substances that offer medicinal or health benefits beyond their nutritional content. The term “nutraceutical” is derived from the combination of “nutrition” and “pharmaceuticals.” These compounds, found in functional foods, contain biologically active molecules that interact with various cellular targets across organ systems. Their effects include antioxidant, anti-inflammatory, anti-proliferative, antimicrobial, and cholesterol-lowering properties [71,72,73]. They can encompass a range of ingredients, including isolated nutrients, phytochemicals, vitamins, minerals, amino acids, fatty acids, herbal remedies, and even genetically modified foods. Research has shown that regular intake of nutraceuticals as part of a diverse diet can help lower the risk of diseases related to oxidative stress, such as Alzheimer’s, cardiovascular diseases, diabetes, and obesity [55,74,75,76].

#### 5.1.1. Antioxidant Nutraceuticals

Dietary antioxidants work synergistically with endogenous antioxidants to neutralize ROS and RNS. Their deficiency is associated with chronic and degenerative diseases. These compounds are structurally diverse and include a range of various molecules.

#### 5.1.2. Ascorbic Acid (Vitamin C)

Ascorbic acid, commonly known as vitamin C, is an essential water-soluble antioxidant that humans cannot synthesize endogenously, unlike most animals and plants, making its dietary intake essential [77]. Rich dietary sources of vitamin C include citrus fruits such as lemons and oranges, as well as parsley and various edible flowers.

Biologically, it plays a crucial role in normal growth and development and is involved in multiple physiological functions [78]. It is characterized as one of the primary antioxidants, directly scavenging ROS and RNS, including superoxide anions, hydroxyl radicals, alkoxyl radicals, hydrogen peroxide, and singlet oxygen [79].

Vitamin C also regenerates other antioxidants such as vitamin E, contributes to collagen biosynthesis, improves iron absorption, and modulates cellular signal transduction. Donating hydrogen reduces tocopherol radicals, thereby restoring the antioxidant function of tocopherol and protecting membrane lipids. Moreover, it reduces the Fenton reaction by neutralizing metal ions that would otherwise promote the formation of free radicals. In plants, ascorbic acid serves as a substrate for ascorbate peroxidase (APX), playing a crucial role in resistance to oxidative stress [80].

#### 5.1.3. Tocopherols (Vitamin E)

It is a group of phenolic compounds containing eight related tocopherols and tocotrienols known as vitamin E. It is a family of lipid-soluble vitamins. The most famous compound of this class is alpha-tocopherol, which has the highest bioavailability, allowing the body to absorb and metabolize it easily. The primary function of alpha-tocopherol is to protect membranes from oxidation, as it inhibits lipid peroxidation by interacting with lipid radicals generated from peroxynitrite and inflammatory reactions [81]. Moreover, tocopherols react with alkyl or alkyl peroxyl radicals by donating hydrogen and in turn produce a tocopherol radical, while ascorbic acid regenerates the reduced form of tocopherol. At low peroxyl radical concentrations and high tocopherol radical concentrations, the tocopherol radical can take hydrogen from the lipid substrate, thereby generating lipid radicals and regenerating tocopherol. In this scenario, the lipid radical contributes to lipid oxidation by interacting with triplet oxygen; thus, the tocopherol radicals may act as prooxidants instead of antioxidants [82,83].

#### 5.1.4. Carotenoids

Carotenoids fall under a large group of molecules called terpenes that are formed by isoprene chains with conjugated double bonds and are fat-soluble [84]. They are divided into two classes based on their structure: simple polyene hydrocarbons, such as β-carotene and lycopene, and xanthophylls, which contain oxidizing groups, including hydroxy, oxo, or epoxy. The broad interest in carotenoids is due to their roles as antioxidants. They act as ROS and RNS scavengers or singlet-oxygen quenchers during lipid oxidation through physical and chemical processes. Tocopherols, known as powerful lipid antioxidants, enhance the presence of carotenoids in the lipid phase of membranes. Their antioxidant activity depends on scavenging singlet oxygen, peroxyl radicals, sulfonyl, thiyl, and NO_2_ radicals, which leads to protection against hydroxyl attack and superoxide radicals.

Carotenoids are also known for their ability to reduce radicals by donating electrons and hydrogen. They form when hydrogen atoms are donated to lipid peroxyl radicals. These radicals react with lipid peroxyl radicals, leading to the formation of non-radical carotene peroxides. Carotene radicals are also prone to molecular-oxygen addition and subsequent reaction with another carotene molecule, producing carotene epoxides and carbonylated carotene compounds. One of the most stable carotene radicals is the β-carotene radical because of the delocalization of its unpaired electron within the conjugated polyene structure [85]. Carotene is a key member of the carotenoid family that can be converted into active vitamin A and into zeaxanthin, which gives the retina its yellow pigment. It has been reported that carotenoids can inhibit cell renewal and transformation, as well as regulate gene expression involved in the development of certain cancers [86].

#### 5.1.5. Phenolics and Flavonoids

Plant polyphenols form a broad class of phenolic compounds, the most prominent being flavonoids flavones and flavonols. These molecules exhibit strong antioxidant activity by effectively scavenging superoxide anions, nitric oxide, and hydroxyl radicals [87]. In edible fruits and vegetables, they also regulate redox-related enzymes, reducing cellular free-radical levels, lowering hyperglycemia, and exerting antiproliferative effects. For phenolic acids, antioxidant strength increases with both the number and the ring position of hydroxyl groups.

Once incorporated into biological membranes, polyphenols insert into the hydrophobic core and slow the oxidation of lipids and proteins; membrane-embedded flavonoids further protect bilayer integrity by physically blocking the entry of ROS and RNS. Dietary sources such as green tea, red grape skins (used in red wine), apples, cocoa, ginkgo biloba, soy, turmeric, various berries, onions, and broccoli are all associated with decreased inflammation and oxidative stress [88].

## 6. Fatty Acids as Modulators of Antioxidant and Anti-Inflammatory Pathways

Fatty acids exert important regulatory effects on oxidative stress and inflammation through multiple mechanisms [89], including activation of nuclear receptors (peroxisome proliferator–activated receptors, PPARs), engagement of membrane G-protein–coupled receptors (GPR120), enzymatic conversion into specialized pro-resolving mediators (SPMs), and formation of bioactive electrophilic derivatives such as nitro-fatty acids. Collectively, these actions converge to inhibit NF-κB–driven pro-inflammatory signaling while enhancing antioxidant defenses [90].

Long-chain polyunsaturated fatty acids (PUFAs), especially omega-3 PUFAs such as eicosapentaenoic acid (EPA) and docosahexaenoic acid (DHA), serve as natural ligands for PPARα, PPARγ, and PPARδ. Upon activation, PPARs heterodimerize with retinoid X receptor (RXR) and regulate target genes involved in lipid metabolism and redox control. In addition, PPARs exert anti-inflammatory effects by trans-repression of NF-κB signaling. Mechanisms include sequestration of coactivators such as CBP/p300, stabilization of the NF-κB inhibitor IκBα, and recruitment of corepressors to NF-κB target promoters. Experimental studies show that oxidized omega-3 fatty acids inhibit NF-κB activation via a PPARα-dependent mechanism, underscoring their dual role in lipid and inflammatory homeostasis [91].

Moreover, PPAR activation complements NRF2 signaling. Both pathways work together to increase antioxidant enzymes, including heme oxygenase-1 (HO-1), NAD(P)H: quinone oxidoreductase 1 (NQO1), and glutamate-cysteine ligase. Studies in metabolic and toxicological models indicate that co-activation of PPARγ and NRF2 provides synergistic protection against oxidative damage, thereby reducing secondary NF-κB activation [92].

Omega-3 PUFAs also activate GPR120 (also known as FFAR4), a membrane receptor highly expressed in macrophages and adipocytes. GPR120 signaling inhibits NF-κB and NLRP3 inflammasome activation while promoting a shift toward an anti-inflammatory M2 macrophage phenotype [93]. In parallel, enzymatic conversion of EPA and DHA produce SPMs such as resolvins, protectins, and maresins, which actively promote resolution of inflammation and limit the oxidative injury [94].

Electrophilic derivatives of unsaturated fatty acids, such as nitro-oleic acid and other nitro-fatty acids, activate both PPARs and NRF2 by covalently modifying cysteine residues in Keap1. This results in the upregulation of antioxidant genes and inhibition of NF-κB signaling, highlighting a direct link between lipid electrophiles and cytoprotective transcriptional responses [95].

Clinical data support the anti-inflammatory potential of omega-3 PUFAs, though outcomes vary by formulation and population. A randomized dose–response trial demonstrated that EPA and DHA supplementation modulate innate immune responses and reduce ex vivo endotoxin-stimulated cytokine release [96]. Large cardiovascular outcome trials produced divergent results: REDUCE-IT showed that high-dose Icosapent ethyl (EPA-only) significantly reduced cardiovascular events, whereas the STRENGTH trial using combined EPA + DHA failed to demonstrate benefit [97,98,99]. These findings suggest that clinical efficacy depends on the specific fatty acid formulation, dosage, and patient context [99].

## 7. Synthetic and Pharmacological Antioxidants: Mechanisms of Action

Synthetic and pharmacological antioxidants have been developed to imitate or enhance the body’s natural antioxidant defenses, especially in cases of diseases related to oxidative stress [100]. These compounds include well-studied agents such as N-acetylcysteine (NAC), edaravone, tempol, and synthetic versions of natural antioxidants, including Trolox, a vitamin E analog. These molecules are used clinically or experimentally to reduce oxidative damage, particularly in conditions like neurodegenerative disorders, ischemia–reperfusion injury, and inflammation-related diseases [100].

The mechanisms of action of these antioxidants are complex. One primary mechanism is free radical scavenging, in which these compounds donate electrons to neutralize ROS and RNS, thereby preventing lipid peroxidation and DNA damage [16]. Studies have demonstrated that ROS acts as a direct scavenger and also serves as a precursor to glutathione, a major endogenous antioxidant [8]. In addition to scavenging, some synthetic antioxidants modulate antioxidant enzyme systems. For example, molecules like tempol and specific pharmacological activators stimulate the expression or activity of SOD and catalase, thereby helping to maintain cellular redox balance [101].

Furthermore, recent studies have shown that certain pharmacological antioxidants exert their effects by modulating gene expression. Compounds such as dimethyl fumarate and resveratrol analogs activate the Nrf2 pathway, a transcription factor that upregulates ARE-driven genes, including those encoding for glutathione peroxidase and NAD(P)H: quinone oxidoreductase [102]. This gene-regulatory mechanism provides a more sustained antioxidant effect by enhancing the body’s defense capacity at the transcriptional level.

These diverse mechanisms underscore the potential of synthetic and pharmacological antioxidants, not only as direct radical scavengers but also as regulators of cellular antioxidant systems and gene expression pathways.

## 8. Anti-Inflammatory Agents with Antioxidant Properties

In recent years, scientific interest has grown in bioactive phytochemicals that have both anti-inflammatory and antioxidant properties. Notably, compounds like curcumin, resveratrol, and quercetin have gained significant attention for their multiple effects in regulating immune responses and redox homeostasis [103].

Curcumin, a polyphenol from Curcuma longa, is well-known for its diverse biological effects. It serves as a potent scavenger of ROS and RNS, thereby decreasing cellular oxidative damage. At the same time, curcumin reduces pro-inflammatory cytokines, such as TNF-α, IL-6, and IL-1β, primarily by inhibiting the NF-κB signaling pathway. This inhibition lessens the activation of inflammation-related genes and helps regulate the immune response to harmful stimuli. Additionally, curcumin stimulates the Nrf2 pathway, promoting the production of protective enzymes such as heme oxygenase-1 (HO-1) and glutathione S-transferases (GSTs), which play crucial roles in mitigating oxidative stress [104].

Resveratrol, a stilbene compound predominantly found in grapes and red wine, exhibits similar dual activities. In vitro and in vivo studies have demonstrated that resveratrol reduces oxidative stress by enhancing the activity of SOD, CAT, and GPx [105,106]. Its anti-inflammatory effects are primarily attributed to inhibition of cyclooxygenase-2 (COX-2) and inducible nitric oxide synthase (iNOS), as well as attenuation of NF-κB signaling. Importantly, resveratrol also influences the behavior of immune cells, including the suppression of T-cell proliferation and promotion of regulatory T-cells (Treg) differentiation, thereby maintaining immune homeostasis [107].

Quercetin, a flavonoid abundantly present in onions, apples, and tea, also exerts dual functionality by influencing oxidative and inflammatory pathways. It effectively inhibits lipid peroxidation and ROS generation, while concurrently suppressing the expression of key inflammatory mediators. Similar to curcumin and resveratrol, quercetin inhibits the NF-κB pathway and downregulates MAPK (mitogen-activated protein kinase) signaling cascades, including ERK, JNK, and p38, leading to reduced production of pro-inflammatory cytokines [108]. Moreover, quercetin enhances mitochondrial function and biogenesis through activation of AMP-activated protein kinase (AMPK), a central regulator of energy and oxidative balance in cells [109].

The immunomodulatory effects of these compounds are not restricted to cytokine modulation but also extend to cellular interactions within the immune system. For instance, curcumin inhibits dendritic cell maturation and antigen presentation, while promoting the expansion of Tregs. Resveratrol enhances macrophage polarization toward the anti-inflammatory M2 phenotype [110], thereby contributing to tissue repair and resolution of inflammation. Quercetin inhibits mast cell degranulation and modulate neutrophil infiltration in inflamed tissues [111]. These mechanisms collectively contribute to the restoration of immune balance and attenuation of chronic inflammation.

Furthermore, these dual-function agents interfere with intracellular signaling pathways that control cellular responses to stress and inflammation. The interplay between NF-κB, Nrf2, and MAPK pathways constitutes a central node targeted by these compounds. By inhibiting NF-κB and activating Nrf2, these agents shift the cellular response from a pro-inflammatory to an antioxidative and cytoprotective state. This dual modulation not only reduces inflammatory damage but also improves resilience against oxidative injury.

In conclusion, dual-function compounds such as curcumin, resveratrol, and quercetin show significant potential as therapeutic agents due to their ability to modulate both inflammation and oxidative stress simultaneously. Their influence on key signaling pathways and immune functions highlights the importance of these natural compounds in preventing and managing chronic inflammatory diseases. Future research should focus on improving their bioavailability and investigating the synergistic effects through combination therapy strategies.

## 9. Clinical Applications and Evidence of Antioxidants and Anti-Inflammatory Agents

An increasing number of preclinical and clinical studies highlight the significant therapeutic potential of antioxidants and anti-inflammatory agents in combating diseases mediated by oxidative stress and chronic inflammation. These compounds, particularly polyphenols such as curcumin, resveratrol, and quercetin, are of high interest due to their ability to act on both oxidative and inflammatory pathways. For instance, curcumin demonstrates significant anti-inflammatory effects by inhibiting NF-κB activation and reducing the expression of COX-2, IL-1β, and TNF-α. In animal models of atherosclerosis, curcumin has been shown to decrease oxidative stress markers such as MDA and boost antioxidant enzyme activities, including SOD and catalase [112].

Similarly, resveratrol exerts cardioprotective effects by activating SIRT1 and AMPK, promoting mitochondrial biogenesis, and reducing ROS generation. Preclinical models of myocardial infarction and cerebral ischemia have demonstrated that resveratrol attenuates infarct size, improves neurological function, and inhibits the release of pro-apoptotic factors. Quercetin has been extensively studied for its ability to restore redox balance by modulating glutathione levels, reducing lipid peroxidation, and suppressing inflammatory cytokines via inhibition of MAPK and JAK-STAT signaling pathways [113].

Clinical evidence, although still emerging, reflects many of these beneficial outcomes (Table 3). For example, curcumin supplementation (500 mg twice daily for 8 weeks) in patients with metabolic syndrome led to significant improvements in endothelial-dependent vasodilation, a reduction in C-reactive protein (CRP), and decreased serum levels of IL-6. In another trial, resveratrol was shown to enhance cerebral blood flow and cognitive performance in patients with mild to moderate Alzheimer’s disease, possibly through its vasodilatory and anti-aggregatory effects. Additionally, Quercetin demonstrated efficacy in lowering systolic blood pressure and improving HDL levels in type 2 diabetic patients with hypertension [114].

Beyond curcumin, resveratrol, and quercetin, several other agents with dual antioxidant and anti-inflammatory properties are currently being evaluated in clinical settings (Table 3). Astaxanthin, a potent carotenoid with strong radical-scavenging activity, is under investigation in heart failure patients, where it is expected to modulate both oxidative stress and inflammatory markers. Similarly, ellagic acid, a polyphenol abundant in pomegranates and berries, has demonstrated clinical benefits in patients with irritable bowel syndrome by increasing total antioxidant capacity and reducing markers of IL-6, CRP, and lipid peroxidation, along with improvements in quality of life.

L-carnitine supplementation combined with exercise has also shown synergistic effects in overweight/obese adults, enhancing antioxidant enzyme activity (CAT, SOD) and reducing ROS, MDA, and IL-6 levels, highlighting the importance of lifestyle-drug combinations. Moreover, eriocitrin, a citrus-derived flavonoid, demonstrated significant reductions in glycemia, systemic inflammation, and oxidative stress, while also enhancing GLP-1 secretion in prediabetic individuals.

In addition to nutraceuticals, novel pharmacological agents are emerging. Sonlicromanol (KH176), a redox-modulating drug, has been shown to improve mitochondrial redox balance and mood in patients with mitochondrial disease. In contrast, GC-4419, an SOD mimetic, is currently being tested in oncology for its dual antioxidant and anti-tumor potential. Crisdesalazine and AT-001 are also being investigated for neurodegenerative and brain oxidative stress–related disorders, respectively, underscoring a growing pipeline of clinically oriented redox-active compounds.

Together, these examples demonstrate that while classical polyphenols remain central to antioxidant–anti-inflammatory research, newer compounds, pharmacological agents, and combination approaches are expanding the therapeutic landscape. However, challenges such as poor bioavailability, short half-life, and inter-individual variability remain common obstacles across these agents, necessitating improved formulations and more rigorous translational efforts.

## 10. Challenges and Limitations

Despite promising results, several challenges limit the clinical translation of antioxidant and anti-inflammatory agents. Classical compounds such as curcumin, resveratrol, and quercetin are constrained by poor bioavailability, rapid metabolism, and inconsistent outcomes across clinical trials. Similarly, newer agents face their own barriers. Astaxanthin, although a potent radical scavenger, has also some bioavailability issues, and its long-term cardiovascular safety remains unclear [115]. Ellagic acid shows benefits in irritable bowel syndrome (IBS) patients, but its effects are highly dependent on gut microbiota metabolism, leading to significant variability in response. Additionally, when L-carnitine is combined with exercise, it exhibits synergistic effects; however, its efficacy is confounded by adherence to lifestyle factors. Eriocitrin has shown improvements in glycemia, oxidative stress, and GLP-1 secretion in prediabetic individuals, but evidence remains limited to short-duration studies with small cohorts [116]. Omega-3 fatty acids, widely studied in metabolic and cardiovascular diseases, often require high doses and show variable outcomes, including paradoxical reductions in insulin sensitivity in specific populations [117]. Novel pharmacological agents such as sonlicromanol and GC-4419, though mechanistically compelling, still face uncertainties regarding long-term safety, patient selection, and cost-effectiveness [118], while compounds like crisdesalazine and AT-001 remain at early investigational stages with limited efficacy data [119]. To address these limitations, several promising strategies have emerged. For example, solid lipid nanoparticles and micelles have been used to enhance curcumin’s dissolution rate, stability, and systemic absorption [120,121,122]. Additionally, prodrugs or analogs were used to improve metabolic stability and pharmacokinetics, although these are more commonly explored in preclinical settings [123]. Enhancers like piperine, an alkaloid from black pepper, substantially improved curcumin exposure by inhibiting hepatic and intestinal glucuronidation and efflux transporters [124].

Collectively, these limitations highlight the need for improved delivery systems, precision-medicine strategies, and well-designed long-term clinical trials to establish reliable therapeutic roles for these agents. Beyond pharmacological limitations, individual variability in therapeutic response is also a critical factor affecting clinical outcomes. Genetic polymorphisms in drug-metabolizing enzymes (e.g., CYP450 isoforms), variation in gut microbiota composition, and differing baseline levels of oxidative stress or inflammation may influence the effectiveness and safety of antioxidant interventions [125]. Some individuals, particularly those with elevated oxidative markers, may respond favorably to treatment, whereas others might experience negligible benefits or even adverse pro-oxidant effects. This reinforces the need for personalized antioxidant therapy, where interventions are tailored based on the individual’s genetic, metabolic, and microbiomic profiles. Moreover, the translational gap from preclinical research to clinical practice remains wide. Although many in vitro and animal studies demonstrate robust anti-inflammatory and antioxidant effects, human trials frequently yield mixed or inconclusive results. These discrepancies often arise from differences in disease complexity, inconsistent dosing regimens, heterogeneous patient populations, and varied endpoints. For example, while polyphenol-based treatments may significantly reduce inflammatory cytokines in animal models, equivalent clinical trials may show only marginal improvements in human subjects [126,127,128]. This gap underscores the importance of standardizing clinical trial methodologies, developing predictive biomarkers, and conducting large-scale, long-term studies to assess real-world efficacy.

## 11. Reconciling Discrepancies Between Preclinical and Clinical Outcomes in Oxidative Stress Modulation

Despite the strong evidence that antioxidant interventions are effective in preclinical models, clinical trials often show minimal or inconsistent effects on oxidative stress biomarkers. Several connected factors contribute to this gap in translation. First, inadequate dosing and pharmacokinetic limitations restrict clinical effectiveness. Many trials use doses that do not reach tissue levels comparable to those in animal or in vitro studies. Non-enzymatic antioxidants often react more slowly with reactive species, such as superoxide, at rates significantly lower than those of nitric oxide, which substantially reduces their effectiveness in living organisms [6]. Second, patient heterogeneity and disease complexity contribute to variability in outcomes. Clinical populations include patients at different disease stages, with diverse comorbidities and genetic backgrounds, making uniform responses unlikely. Trials are often underpowered or lack stratified designs to account for this variability [129]. Third, insensitive or non-specific biomarkers can obscure subtle effects. Standard plasma measures, such as MDA, SOD, or 8-isoprostanes, may not capture localized oxidative stress, and methodological inconsistencies further complicate interpretation. Fourth, treatment duration and study design issues play a role. Chronic diseases evolve over decades, whereas most clinical studies are conducted over weeks or months, limiting the ability to detect long-term molecular or clinical benefits [130,131]. Finally, mechanistic discordance between preclinical and clinical settings contributes to outcome discrepancies. Preclinical models often rely on severe oxidative insults or controlled environments, while human disease involves multifactorial, chronic processes. Moreover, some antioxidants exert effects through non-redox pathways, complicating biomarker-based assessments [132]. Together, these considerations emphasize the need for optimized dosing strategies, patient stratification, longer follow-up, and deployment of sensitive biomarker panels to bridge the gap between promising preclinical results and variable clinical outcomes.

## 12. Emerging Strategies and Future Directions

### 12.1. Combination Therapies: Synergistic Targeting of Oxidative Stress and Inflammation

Given the complex interplay between oxidative stress and inflammation in the development of chronic diseases, combination therapies that target both pathways simultaneously have shown promising results. Using antioxidants and anti-inflammatory agents together can create synergistic effects, resulting in increased effectiveness and fewer side effects compared to using either therapy alone. For instance, a combination of N-acetylcysteine (NAC) and curcumin significantly reduced oxidative damage and neuroinflammation in models of Alzheimer’s disease by lowering neuronal apoptosis, lipid peroxidation, and pro-inflammatory cytokines such as TNF-α and IL-1β [133]. Similarly, the co-administration of resveratrol and omega-3 fatty acids has been shown to improve insulin sensitivity and decrease systemic inflammation in metabolic syndrome by modulating key signaling pathways like NF-κB and MAPK [134,135].

### 12.2. Targeted Delivery Systems: Nanotechnology-Driven Precision in Antioxidant and Anti-Inflammatory Therapy

Pharmacokinetic limitations, including poor aqueous solubility, rapid degradation, and nonspecific tissue distribution, often hinder the clinical translation of antioxidant and anti-inflammatory therapies. To address these challenges, nanotechnology-based delivery systems have emerged as a highly promising strategy, offering enhanced bioavailability, sustained release, and targeted delivery of therapeutic agents.

Liposomal delivery systems, composed of biocompatible phospholipid bilayers, have demonstrated strong potential for encapsulating both hydrophilic and lipophilic antioxidants. For example, curcumin-loaded liposomes have been shown to significantly improve therapeutic efficacy in experimental colitis by enhancing mucosal retention and reducing systemic toxicity compared to free curcumin [136].

Polymeric nanoparticles, such as those made from poly(lactic-co-glycolic acid) (PLGA), enable controlled release and protection of unstable bioactive compounds. Quercetin has demonstrated improved solubility, bioavailability, and cellular uptake when delivered via PLGA nanoparticles, resulting in better outcomes in models of hepatic oxidative stress [137].

Solid lipid nanoparticles (SLNs) and nanostructured lipid carriers (NLCs) are lipid-based nanocarriers offering enhanced encapsulation efficiency, stability, and scalability. SLNs loaded with resveratrol were found to effectively protect neuronal cells from ROS-induced apoptosis by facilitating mitochondrial delivery and sustained intracellular release [138]. Emerging stimuli-responsive nanocarriers, designed to respond to specific pathological environments such as oxidative stress or acidic pH, provide even greater precision. For instance, glutathione-sensitive nanoparticles have been engineered to release their antioxidant cargo selectively in cells exhibiting high oxidative stress, thereby minimizing damage to healthy tissues [139]. These advances in nanotechnology enhance the therapeutic index of antioxidant and anti-inflammatory agents, offering a new frontier for precision medicine in managing chronic diseases associated with oxidative stress and inflammation.

### 12.3. Personalized Medicine and Biomarkers: Toward Precision Redox Therapy

Advancements in personalized medicine have opened new ways to customize antioxidant and anti-inflammatory treatments based on individual biomarker profiles. Biomarkers such as 8-hydroxy-2′-deoxyguanosine (8-OHdG), malondialdehyde (MDA), C-reactive protein (CRP), IL-6, and TNF-α serve as indicators of oxidative damage and inflammatory status. Multi-omics technologies now enable the integration of genomic, transcriptomic, and metabolomic data to guide personalized therapy. Elevated levels of urinary 8-OHdG, for example, have been used to predict oxidative DNA damage and assess response to antioxidant therapies for cardiovascular patients. The combination of biomarker monitoring and computational tools can further refine treatment protocols, leading to more effective and individualized therapies [140,141,142].

### 12.4. Gut Microbiome as a Modulator of Oxidative Stress and Inflammation

Recent advances identify the gut microbiome as a promising therapeutic target to break the cycle of oxidative stress and inflammation in chronic and metabolic diseases. Dysbiosis is associated with the excessive production of pro-oxidant metabolites, weakened antioxidant defenses, and the release of microbial products, such as lipopolysaccharides, that activate the NF-κB and inflammasome pathways [143]. These changes promote systemic inflammation and oxidative damage, contributing to conditions like cardiovascular disorders, type 2 diabetes, neurodegeneration, and cancer. Interventions that restore microbial balance are an emerging strategy to counteract these processes. Probiotics and prebiotics have demonstrated effectiveness in reducing circulating markers of oxidative stress and inflammation in metabolic syndrome and diabetes [144,145]. Dietary polyphenols not only boost microbial diversity but also work with microbial metabolism to produce metabolites that activate NRF2 signaling and reduce NF-κB activity [47,146]. Additional experimental approaches, such as fecal microbiota transplantation, have shown promise in improving insulin sensitivity and reducing low-grade inflammation in metabolic disorders [147,148,149].

Positioning the microbiome as a therapeutic target opens novel opportunities to complement existing antioxidant and anti-inflammatory therapies. By modulating host redox balance and immune signaling at their source, microbiome-directed interventions may offer a durable and systemic way to break the vicious cycle of oxidative stress and inflammation.

### 12.5. Synthetic Derivatives of Antioxidants: Advances and Therapeutic Potential

Advancements in medicinal chemistry have led to the development of synthetic derivatives of natural antioxidants (Figure 3), aiming to overcome limitations such as poor bioavailability, rapid metabolism, and limited stability. To enhance the therapeutic potential of natural antioxidants, researchers have developed various synthetic derivatives aimed at improving bioavailability, stability, and efficacy. These modifications have resulted in compounds with superior antioxidant, anti-inflammatory, and anticancer properties compared to their natural counterparts. For example, Curcumin has demonstrated multiple bioactivities. However, its clinical application is limited due to poor bioavailability and rapid metabolism. To address these limitations, several curcumin analogs have been synthesized. EF24 (3,5-bis[(2-fluorophenyl)methylidene]piperidin-4-one): EF24 is a synthetic analog of curcumin that exhibits enhanced bioavailability and potent bioactivity, including anti-cancer, anti-inflammatory, and anti-bacterial effects [150]. It induces cell cycle arrest and apoptosis via a redox-dependent mechanism, primarily through its inhibitory effect on the NF-κB pathway. Recent studies have reported the synthesis of twelve asymmetric curcumin analogs using Pabon’s method [151]. These compounds have shown improved antioxidant abilities compared to curcumin, with some exhibiting low to moderate yields. Resveratrol Derivatives have been extensively studied for their antioxidant properties. To enhance its efficacy, several synthetic derivatives have been developed. Various resveratrol derivatives have been synthesized to improve their biological activities, including anticancer, antioxidant, and anti-inflammatory effects. These modifications aim to overcome the limitations of resveratrol’s bioavailability and stability [152]. Flavonoid-based synthetic antioxidants possess significant antioxidant properties. Synthetic modifications of flavonoids have led to compounds with enhanced stability and bioactivity [153]. Synthesis of various flavonoid derivatives has been explored to improve their antioxidant and therapeutic properties. These derivatives have been synthesized to enhance the pharmacological activities of flavonoids while overcoming their limitations. These synthetic derivatives represent promising strategies to increase the therapeutic potential of natural antioxidants, addressing challenges such as poor bioavailability and rapid metabolism. Additionally, other derivatives, including sulfonamide-modified gallic acid and Trolox-based compounds, exhibit enhanced antioxidant and anti-inflammatory properties, offering improved therapeutic potential [154]. For instance, gallic acid sulfonamide derivatives demonstrate increased oral bioavailability and sustained antioxidant activity, while Trolox derivatives have shown promising anti-inflammatory effects [154,155]. Such modifications not only enhance the pharmacokinetic profiles of these compounds but also expand their therapeutic applications in treating disorders related to oxidative stress. The ongoing research on these synthetic derivatives highlights their important role in developing new therapeutic agents that target oxidative stress and inflammation.

## 13. Conclusions

This review has explored the complex link between oxidative stress and inflammation, highlighting their key roles in the development and progression of various chronic diseases. The evidence shows that oxidative stress not only causes cellular damage but also worsens inflammatory responses, sustaining pathological processes in conditions such as cardiovascular disease, neurodegeneration, chronic kidney disease, diabetes, and cancer. Natural and synthetic antioxidants are promising in reducing these harmful effects by neutralizing reactive oxygen species and affecting pro-inflammatory pathways. Importantly, combination therapies that include antioxidant and anti-inflammatory agents demonstrate synergistic effects, providing better results and fewer side effects than using single drugs. Recent advances in nanotechnology and targeted delivery systems have significantly improved the pharmacokinetics of these agents, enabling precise, site-specific treatments with minimal systemic side effects. The development of personalized medicine and the use of biomarkers have further enhanced therapeutic strategies, allowing for tailored treatments based on individual oxidative and inflammatory profiles. This move toward precision medicine offers great hope for improved clinical outcomes and a reduction in the burden of chronic diseases associated with oxidative stress.

## Figures and Tables

**Figure 1 diseases-13-00309-f001:**
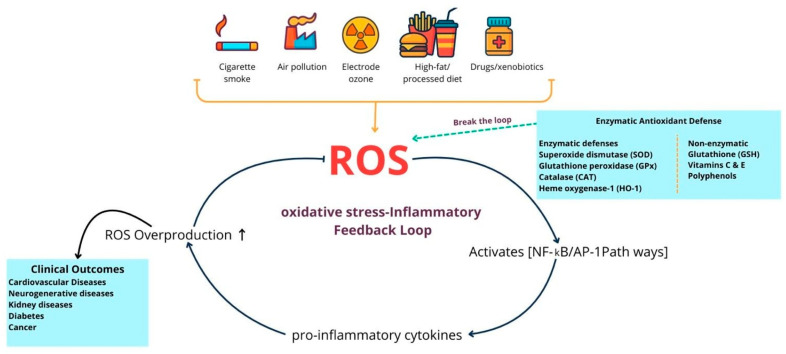
Endogenous oxidative stress and antioxidant defenses in human diseases. This diagram illustrates the balance between external sources of oxidative stress, such as smoking, pollutants, radiation, diet, xenobiotics, and internal sources like chronic inflammation, along with the antioxidant defenses that mitigate their effects. Enzymatic systems, such as superoxide dismutase, catalase, glutathione peroxidase, and heme oxygenase-1, along with non-enzymatic molecules including glutathione, vitamins C and E, and polyphenols, neutralize reactive oxygen species (ROS) and protect macromolecules from damage. When ROS production exceeds antioxidant capacity, it results in oxidative damage to lipids, proteins, and DNA, which in turn increases inflammatory signaling, leading to further ROS production and creating a self-perpetuating cycle of oxidative stress and inflammation. Antioxidant defenses aim to break this cycle and prevent ROS-driven inflammation from escalating. If this balance is not restored, persistent oxidative stress and cellular damage occur, contributing to the development and progression of cardiovascular disease, type 2 diabetes mellitus, neurodegenerative disorders, chronic kidney disease, and cancer. This illustration was created using Canva (https://www.canva.com, accessed on 12 September 2025).

**Figure 2 diseases-13-00309-f002:**
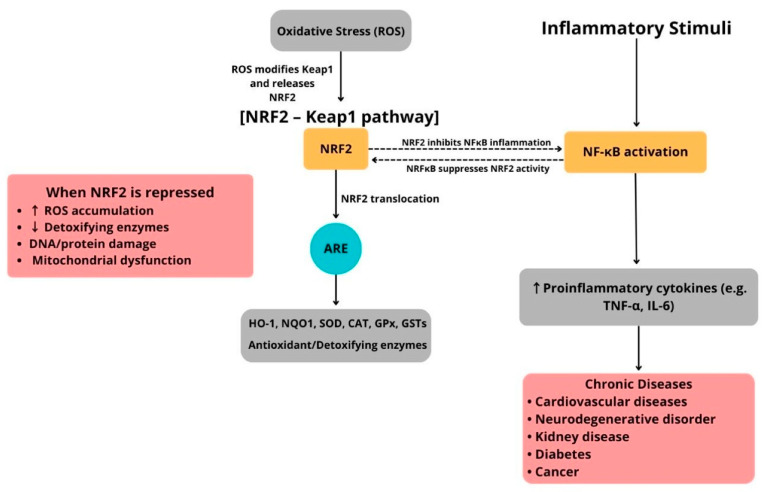
Schematic illustration of the NRF2–Keap1/ARE pathway and its interaction with NF-κB–mediated inflammation in human diseases.

**Figure 3 diseases-13-00309-f003:**
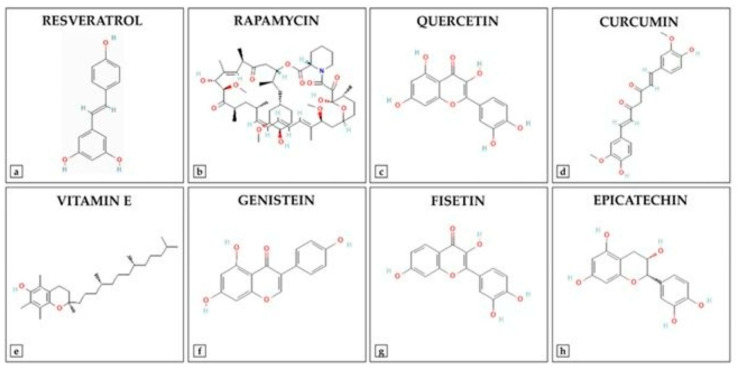
Chemical Structures of Key Antioxidant Compounds.

**Table 2 diseases-13-00309-t002:** Comparison Between Endogenous Enzymatic and Non-Enzymatic Antioxidants.

Category	Enzymatic Antioxidants	Non-Enzymatic Antioxidants
Definition	Enzymes produced by the body catalytically neutralize reactive oxygen and nitrogen species (ROS/RNS)	Small, naturally occurring molecules that directly interact with and neutralize free radicals
Mechanism of Action	Convert ROS/RNS into less harmful substances through stepwise enzymatic reactions	Directly scavenge and deactivate ROS/RNS or regenerate other antioxidant molecules
Key Examples	Superoxide dismutase (SOD) Catalase (CAT) Glutathione peroxidase (GPx	Glutathione (GSH) Uric acid Melatonin Alpha-lipoic acid
Localization	SOD1: Cytosol SOD2: Mitochondria SOD3: Extracellular CAT: Liver, kidneys, RBCs GPx: Mitochondria and cytosol	GSH: Cytosol and mitochondria Uric acid: Plasma Melatonin: All cellular compartments Lipoic acid: Both intra- and extracellular
Distinct Features	Require metal cofactors (e.g., Cu, Zn, Mn, Fe) Function as an integrated antioxidant system	React with multiple ROS types Capable of crossing membranes (e.g., melatonin) Support vitamin regeneration
Additional Functions	Decompose hydrogen peroxide Protect proteins, DNA, and lipids from oxidative damage	Regulate circadian rhythm (melatonin) Detoxify peroxynitrite and hydroxyl radicals Maintain redox balance

**Table 3 diseases-13-00309-t003:** Examples of Clinical Trials Investigating Anti-Inflammatory and Antioxidant Therapies Targeting Oxidative Stress and Inflammation.

Compound/Drug	Condition/ Population	Phase & Design	Dose	Duration	Key Outcomes	Trial ID	Year
Antioxidant cocktail (Vitamins E/C + Tocopherol, Ascorbic Acid, Selenium)	Overweight children	RCT, placebo-controlled	TP,400 IU; 500 mg, SE 500 mg, AA 50 µg daily	4 months	Reduction of 8-iso-PGF_2_α; ↑ GPx and ↑ SOD activity	NCT01316081	2011
Astaxanthin	Heart failure patients	RCT (protocol stage)	20 mg of astaxanthin per day	8 weeks	No results posted	IRCT20200429047235N3	2024
Resveratrol	Healthy adult smokers	Phase III, crossover RCT	500 mg/day	30 days per arm (3 months)	No results posted	NCT01492114	2012
Curcuminoids	Hemodialysis patients	RCT, double-blind, placebo-controlled	500 mg/8 h	12 weeks	No posted results yet	NCT06829186	2025
Ellagic Acid (polyphenol)	IBS patients	RCT	180 mg of EA per day	8 weeks	Antioxidant index improved. ↑ TAC; ↓ MDA	IRCT20141025019669N11	2019
L-carnitine + exercise	Overweight/obese adults	RCT	1 g/day + exercise	12 weeks	Increase in CAT and SOD; Decrease in ROS, MDA, and IL-6	NR	NR
ω-3 fatty acids	Type 2 diabetes patients	RCT	marine n-3 fatty acids in 100 mL Omegaven (1.25–2.82 g EPA + 1.44–3.09 g DHA)	9 weeks	Reduced insulin sensitivity and altered proportion of carbohydrate vs. fat oxidation No enzyme/cytokine results posted on the registry	NCT00829569	2011
Eriocitrin (Eriomin)	Prediabetic individuals	Crossover RCT	200 mg/day	12 weeks	Reduced glycemia, systemic inflammation, and oxidative stress, and increased GLP1 No results for cytokines	NCT03928249	2020
Crisdesalazine	Neurodegenerative disorders	Phase I (ongoing)	Not disclosed	NS	Free-radical scavenger; safety & PK data	NR	NR
Sonlicromanol (KH176)	Mitochondrial disease patients	Phase II RCT	50 mg twice daily	1 month	Improved safety, mood, mitochondrial redox balance	NCT04165239	2022
GC-4419 (SOD mimetic)	Head & Neck cancer	Phase I dose escalation	15–170 mg	NS	No data for plasma SOD/GPx/CAT	NCT01921426	2013
AT-001	Brain oxidative stress	Phase I	NS	12 weeks	no published enzyme/cytokine panel	NCT01731093	2012
Lutein supplementation	Healthy adult nonsmokers	RCT	200 mg	12 weeks	No data for SOD/GPx/CAT No IL-6/TNF-α/IL-1β results posted on the registry	NCT01056094	2010
Oxytocin	Healthy adult	Phase II	48 IU intranasal 4×/day (QID)	NS	No results posted for SOD/GPx/CAT No cytokine results posted	NCT04732247	2022

NR: No registered clinical trial found based on secondary literature or an unregistered study. NS: Not specified. RCT, randomized clinical trials; AA, Ascorbic Acid; TE, Tocopherol Equivalent; SE, Selenium; 8-iso-PGF_2_α, 8-isoprostaglandin F2α; GPx, Glutathione Peroxidase; SOD, Superoxide Dismutase; CAT, Catalase; TAC, Total Antioxidant Capacity; MDA, Malondialdehyde; IL-6, Interleukin-6; TNF-α, Tumor Necrosis Factor-alpha; IL-1β, Interleukin-1 beta; QID, Quarter in Die (four times daily); GLP-1, Glucagon-Like Peptide-1.

## Data Availability

Not applicable.

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
