# Peer review of "Exploring the Interplay of Antioxidants, Inflammation, and Oxidative Stress: Mechanisms, Therapeutic Potential, and Clinical Implications"

_diseases, 2025, doi:10.3390/diseases13090309_

Round 1
Reviewer 1 Report
Comments and Suggestions for Authors
The manuscript is well-organized and well-written. However, it may be significantly improved, including new chapters (sections) regarding the important role of antioxidants (natural or synthetic) in organ protection (heart, kidney, gastrointestinal tract, and brain). Many papers and reviews have examined this topic. Therefore, the authors should add this information.
Comments on the Quality of English Language
The manuscript contains some typing errors. I suggest a careful evaluation by an English mother language reader.
Author Response
Comments and Suggestions for Authors
The manuscript is well-organized and well-written. However, it may be significantly improved, including new chapters (sections) regarding the important role of antioxidants (natural or synthetic) in organ protection (heart, kidney, gastrointestinal tract, and brain). Many papers and reviews have examined this topic. Therefore, the authors should add this information.
Response:
We thank the reviewer for this valuable suggestion. We agree that the role of antioxidants in organ protection is highly relevant and complements our discussion. In the revised manuscript, we have added a dedicated subsection discussing the protective effects of natural and synthetic antioxidants on major organs, including the heart, kidney, gastrointestinal tract, and brain. We believe this addition broadens the clinical and mechanistic perspective of our work and significantly strengthens the manuscript.
Comments on the Quality of English Language: The manuscript contains some typing errors. I suggest a careful evaluation by an English mother language reader.
The manuscript contains some typing errors. I suggest a careful evaluation by an English mother language reader.
Response:
We appreciate this comment. Grammatical and typographical errors have been fixed to improve overall readability.
Reviewer 2 Report
Comments and Suggestions for Authors
Congratulations to the authors for this comprehensive and timely review, which adeptly bridges fundamental mechanisms of oxidative stress and inflammation with their clinical implications. The manuscript stands out for its interdisciplinary approach, integrating biochemistry, pharmacology, and translational medicine, and for its up-to-date references, including emerging topics like nanotechnology and personalized therapies. The inclusion of clinical trial data (e.g., Table 3) and the discussion of nutraceuticals add significant practical value.
Below are specific comments to further strengthen the manuscript:
- The subsections under "Oxidative Stress" (e.g., 2.1.1.1–2.1.1.4) are highly fragmented. Consolidating or reorganizing these would improve readability. For example, merging related ROS/RNS species (e.g., superoxide and hydroxyl radicals) under broader headings could enhance clarity.
- Key figures (e.g., Figure 1, 2) are referenced but not provided in the submitted PDF. These should be included with detailed legends to illustrate critical concepts like the ROS-inflammation loop or antioxidant mechanisms.
- While the poor bioavailability of compounds like curcumin is noted, a dedicated paragraph on strategies to overcome this (e.g., nanoformulations, prodrugs) would add depth. Highlighting specific examples (e.g., liposomal curcumin in Table 3) would reinforce this point.
- Address discrepancies in clinical outcomes more explicitly. For instance, why do some trials report minimal changes in oxidative stress biomarkers despite strong preclinical data? Discussing factors like dosing, patient heterogeneity, or biomarker sensitivity could provide nuance.
- Expand the bidirectional ROS-inflammation loop (Section 3) with a schematic to visualize key pathways (NF-κB, Nrf2). Additionally, clarify how high-dose antioxidants might act as pro-oxidants and the implications for therapy.
- Correct minor errors (e.g., "polymorphi" → "polymorphic" in Section 7).
- Standardize abbreviations (e.g., define "MDA" before use in Table 3).
- Reduce redundancy (e.g., repetitive mentions of nutraceuticals in the Introduction/Conclusions).
- The "Emerging Strategies" section is a strength but could be enhanced with specific examples of ongoing trials (e.g., NCT numbers) for combination therapies or targeted delivery systems (10.3390/pharmaceutics17010114 ; 10.3390/molecules30030653).
- Consider adding a brief discussion on the gut microbiome’s role in modulating oxidative stress/inflammation, given its growing relevance in chronic diseases.
Author Response
Congratulations to the authors for this comprehensive and timely review, which adeptly bridges fundamental mechanisms of oxidative stress and inflammation with their clinical implications. The manuscript stands out for its interdisciplinary approach, integrating biochemistry, pharmacology, and translational medicine, and for its up-to-date references, including emerging topics like nanotechnology and personalized therapies. The inclusion of clinical trial data (e.g., Table 3) and the discussion of nutraceuticals add significant practical value.
Below are specific comments to further strengthen the manuscript:
- The subsections under "Oxidative Stress" (e.g., 2.1.1.1–2.1.1.4) are highly fragmented. Consolidating or reorganizing these would improve readability. For example, merging related ROS/RNS species (e.g., superoxide and hydroxyl radicals) under broader headings could enhance clarity.
We appreciate the reviewer’s insightful suggestion regarding the structure of the subsections under “Oxidative Stress.” We agree that high fragmentation may reduce readability. Therefore, we have reorganized this section by combining related reactive oxygen and nitrogen species (ROS/RNS).
- Key figures (e.g., Figure 1, 2) are referenced but not provided in the submitted PDF. These should be included with detailed legends to illustrate critical concepts like the ROS-inflammation loop or antioxidant mechanisms.
We thank the reviewer for pointing this out. The omission of Figures 1 and 2 in the submitted PDF was unintentional. In the revised version, we have included these key figures (revised) along with detailed legends.
- While the poor bioavailability of compounds like curcumin is noted, a dedicated paragraph on strategies to overcome this (e.g., nanoformulations, prodrugs) would add depth. Highlighting specific examples (e.g., liposomal curcumin in Table 3) would reinforce this point.
We appreciate the reviewer’s valuable suggestion. In the revised manuscript, we have added a dedicated paragraph discussing strategies to overcome the poor bioavailability of natural antioxidants such as curcumin. Specifically, we highlight the role of nanoformulations (e.g., liposomal and polymeric nanoparticles), structural modifications (e.g., prodrugs and analogs), and combination therapies with bioavailability enhancers (e.g., piperine). To reinforce this point, we have also included specific examples such as liposomal curcumin and curcumin–piperine formulations, which have demonstrated improved pharmacokinetics and therapeutic efficacy.
- Address discrepancies in clinical outcomes more explicitly. For instance, why do some trials report minimal changes in oxidative stress biomarkers despite strong preclinical data? Discussing factors like dosing, patient heterogeneity, or biomarker sensitivity could provide nuance.
We thank the reviewer for this important observation. In the revised manuscript, we have expanded the discussion to more clearly address the differences between preclinical and clinical outcomes in antioxidant interventions. Specifically, we note that while preclinical models often demonstrate significant reductions in oxidative stress, clinical trials sometimes reveal minimal or inconsistent changes in oxidative stress biomarkers. Possible explanations were discussed. By discussing these limitations, we aim to offer a more nuanced understanding of the translational gap and emphasize the need for better biomarker panels and personalized approaches in future clinical studies.
- Expand the bidirectional ROS-inflammation loop (Section 3) with a schematic to visualize key pathways (NF-κB, Nrf2). Additionally, clarify how high-dose antioxidants might act as pro-oxidants and the implications for therapy.
- Correct minor errors (e.g., "polymorphi" → "polymorphic" in Section 7).
- Standardize abbreviations (e.g., define "MDA" before use in Table 3).
- Reduce redundancy (e.g., repetitive mentions of nutraceuticals in the Introduction/Conclusions).
We thank the reviewer for these valuable suggestions.
- Bidirectional ROS–inflammation loop: We have expanded this section to include a schematic (Fig. 2) illustrating the link between oxidative stress and inflammation. The figure now highlights activation of NF-κB, which promotes pro-inflammatory cytokine production, and Nrf2, which regulates antioxidant gene expression.
- High-dose antioxidants as pro-oxidants: We have added a discussion clarifying that at high concentrations, certain antioxidants (e.g., vitamin C, polyphenols) can exhibit pro-oxidant activity by generating free radicals or interacting with transition metals.
- The "Emerging Strategies" section is a strength but could be enhanced with specific examples of ongoing trials (e.g., NCT numbers) for combination therapies or targeted delivery systems (10.3390/pharmaceutics17010114 ; 10.3390/molecules30030653).
Response:
We thank the reviewer for highlighting the “Emerging Strategies” section. As of the latest registry checks, there are no posted primary-endpoint results or peer-reviewed outcome publications for either identifier on ClinicalTrials.gov.
- Consider adding a brief discussion on the gut microbiome’s role in modulating oxidative stress/inflammation, given its growing relevance in chronic diseases.
Response:
We thank the reviewer for emphasizing the importance of the gut microbiome. In the revised manuscript, we have included a brief discussion on its role in modulating oxidative stress (inflammation under the emerging strategies section).
- Minor Corrections and Standardizations
- Corrected “polymorphi” to “polymorphic” in Section 7.
- Defined all abbreviations at first mention (e.g., “MDA (malondialdehyde)”) and standardized them across text and tables.
- Reduced repetitive references to nutraceuticals in the Introduction and Conclusion to eliminate redundancy.
Reviewer 3 Report
Comments and Suggestions for Authors
Although this manuscript describes a topic discussed also in other reviews, it is suitable for publication since the approach is different and deserves interesting info to study the topic.
In particular, the authors disclosed also the role of antioxidants in clinical practice highlighting their important role.
I would like to suggest the inclusion of chemical structures of compounds analyzed but also the introduction of a new paragraph with the development of synthetic derivatives based on antioxidants.
Author Response
Comments and Suggestions for Authors
Although this manuscript describes a topic discussed also in other reviews, it is suitable for publication since the approach is different and deserves interesting info to study the topic. In particular, the authors disclosed also the role of antioxidants in clinical practice highlighting their important role.
We sincerely thank the reviewer for the positive and encouraging feedback. We are pleased that the distinct approach of our manuscript, particularly the integration of antioxidant mechanisms with their potential clinical applications, was recognized as valuable. In revising the manuscript, we have further emphasized the translational aspects by:
- Expanding the discussion on the clinical implications of antioxidant therapies across different organ systems .
- Clarifying limitations in clinical translation, including bioavailability challenges and interindividual variability in response.
I would like to suggest the inclusion of chemical structures of compounds analyzed but also the introduction of a new paragraph with the development of synthetic derivatives based on antioxidants.
We thank the reviewer for this constructive suggestion. To enhance the manuscript, we have implemented the following changes:
- Chemical Structures of Key Compounds:
A new figure (Figure 3) has been added, showing the chemical structures of major natural antioxidants discussed in the manuscript, including curcumin, resveratrol, quercetin, and vitamin E. - Synthetic Derivatives of Antioxidants:
A new paragraph has been incorporated discussing the development of synthetic derivatives and analogs designed to improve potency, stability, and bioavailability.
Reviewer 4 Report
Comments and Suggestions for Authors
The review titled "Exploring the Interplay of Antioxidants, Inflammation, and Oxidative Stress: Mechanisms, Therapeutic Potential, and Clinical Implications " is an interesting work, but major information missed to conclude.
-The abstract is weak and lacks appeal. You have to clearly highlight:
The central objectives of the review.
The novelty and relevance of the review.
- Lines 50–146 are redundant and poorly structured.
Remove lines 50–125 entirely.
Merge section lines 126-146 1.4. Interrelationships Between Oxidative Stress, Antioxidants, and Inflammation with the initial introduction to create a more cohesive and informative opening.
-The table 1 is inconsistence:
The Production Source column mixes locations (e.g., mitochondria) with reaction mechanisms.
Standardize the column to either sources or biochemical reactions.
Add references in the table
-You have to review the paragraph 198-212 it is not structured
-Figure 1 title placement is incorrect and the figure lacks innovation.
-Move the section “2.3. Impact of Oxidative Stress on Cellular Structures and Functions” before the discussion on antioxidants to maintain logical progression from damage to defense.
-Figure 2: you have to improve the type and design of figures
-The review must coverage the NRF2 key pathway:
Add a dedicated subsection explaining: NRF2-Keap1 pathway, its role in antioxidant gene regulation and Cross-talk with inflammatory pathways (e.g., NF-κB).
-The author must speak about the role of the fatty acids in the antioxidants and anti-inflammatory effects (for example the PPAR (activated by fatty acid) role in inhibition of NF-κB ……. ) and add relevant studies or clinical trials.
-In the table 3 and the part Anti-Inflammatory Agents with Antioxidant Properties and in the challenge and limitations: Only a few compounds discussed (curcumin resveratrol and quercetin) !
- The Key Outcomes column in the table lacks specificity. It is necessary to revise this column to include references to specific enzymes (such as SOD, GPx, and CAT), cytokines (such as IL-6, TNF-α, and IL-1β), and to clearly state quantitative results or observed trends.
also “Oxidative & inflammatory markers assessed” are not a results !
- The consistency and clarity should be maintained throughout the table
-You should develop the clinical section with a sound scientific explanation and focus on the information you want to convey.
The author must show the major revisions made in the text by highlighting the changes in a different colored text.
It is imperative to consider all these remarks to reinforce the manuscript's quality and conclude more accurately.
Author Response
Comments and Suggestions for Authors
The review titled "Exploring the Interplay of Antioxidants, Inflammation, and Oxidative Stress: Mechanisms, Therapeutic Potential, and Clinical Implications " is an interesting work, but major information missed to conclude.
-The abstract is weak and lacks appeal. You have to clearly highlight:
The central objectives of the review.
The novelty and relevance of the review.
We thank the reviewer for this helpful comment. We have revised the abstract to better highlight the main objectives of the review, the uniqueness of our approach, and its clinical and scientific importance. In the updated version, we clearly state the primary goal of the review, point out how our work differs from previous reviews in the field, and emphasize the potential implications of our findings and discussion for future research and clinical practice. We believe these changes significantly enhance the clarity and impact of the abstract.
- Lines 50–146 are redundant and poorly structured.
Remove lines 50–125 entirely.
Merge section lines 126-146 1.4. Interrelationships Between Oxidative Stress, Antioxidants, and Inflammation with the initial introduction to create a more cohesive and informative opening.
We thank the reviewer for this helpful observation. As recommended, we removed lines 50–125, which contained redundant and less organized information. To improve clarity and cohesion, we merged the content from lines 126–146 (Interrelationships Between Oxidative Stress, Antioxidants, and Inflammation) with the beginning of the introduction. This restructuring allowed us to create a more concise, cohesive, and informative opening section that better frames the review's objectives and scope. We believe these revisions have greatly enhanced the manuscript's readability and overall flow.
-The table 1 is inconsistence:
The Production Source column mixes locations (e.g., mitochondria) with reaction mechanisms.
Standardize the column to either sources or biochemical reactions.
Add references in the table
We thank the reviewer for this helpful point. We have standardized and revised Table 1 accordingly. We also added representative literature references for each entry.
-You have to review the paragraph 198-212 it is not structured
We appreciate the reviewer’s feedback. The paragraph from lines 198–212 has been carefully revised to improve its structure, clarity, and logical flow. Redundant statements were removed, and the remaining content was reorganized to ensure a clear progression of ideas. The revised version now presents the concepts in a more coherent and reader-friendly manner, better aligning with the review's objectives.
-Figure 1 title placement is incorrect and the figure lacks innovation.
We thank the reviewer for this observation. The title of Figure 1 has been repositioned according to journal guidelines to ensure proper placement and readability. Furthermore, we have revised the figure to enhance its originality and conceptual clarity.
-Move the section “2.3. Impact of Oxidative Stress on Cellular Structures and Functions” before the discussion on antioxidants to maintain logical progression from damage to defense.
Agreed. We moved section 2.3, 'Impact of Oxidative Stress on Cellular Structures and Functions,' to appear before the discussion on antioxidants. This reorganization guarantees a logical flow from the mechanisms of cellular damage to the subsequent defense systems.
-Figure 2: you have to improve the type and design of figures
We have improved Figure 2 by upgrading both its type and overall design to make it clearer, more visually appealing, and more informative for readers. The updated figure now better illustrates the integrated mechanisms of oxidative stress and antioxidant defenses, highlighting novel aspects such as the cross-talk between NRF2–Keap1 and NF-κB pathways, as well as the role of fatty acids in modulating antioxidant and anti-inflammatory responses.
-The review must coverage the NRF2 key pathway:
Add a dedicated subsection explaining: NRF2-Keap1 pathway, its role in antioxidant gene regulation and Cross-talk with inflammatory pathways (e.g., NF-κB).
Thank you for this excellent suggestion. We have added a new section on the NRF2–Keap1 Pathway and its Cross-Talk with Inflammatory Signaling.
-The author must speak about the role of the fatty acids in the antioxidants and anti-inflammatory effects (for example the PPAR (activated by fatty acid) role in inhibition of NF-κB ……. ) and add relevant studies or clinical trials.
Thank you for this valuable suggestion. We have added a new section on Fatty Acids as Modulators of Antioxidant and Anti-Inflammatory Pathways.
-In the table 3 and the part Anti-Inflammatory Agents with Antioxidant Properties and in the challenge and limitations: Only a few compounds discussed (curcumin resveratrol and quercetin) !
We appreciate the reviewer’s comment. Our aim in Table 3 and the “Anti-Inflammatory Agents with Antioxidant Properties” section was to highlight key, well-studied compounds (curcumin, resveratrol, and quercetin) as representative examples with dual antioxidant and anti-inflammatory effects. However, we agree that adding more compounds would enhance the discussion and provide a broader perspective. Therefore, we have expanded the discussion section to include other agents listed in Table 3 and have revised the challenges and limitations section accordingly.
- The Key Outcomes column in the table lacks specificity. It is necessary to revise this column to include references to specific enzymes (such as SOD, GPx, and CAT), cytokines (such as IL-6, TNF-α, and IL-1β), and to clearly state quantitative results or observed trends. Also “Oxidative & inflammatory markers assessed” are not a results !
Thank you for highlighting this critical issue. Key outcomes have been revised accordingly.
- The consistency and clarity should be maintained throughout the table
Done. Thank you.
-You should develop the clinical section with a sound scientific explanation and focus on the information you want to convey.
We thank the reviewer for this excellent suggestion. In the revised version, we have expanded the clinical section to provide a more comprehensive and scientifically grounded explanation. Specifically, we now emphasize the mechanistic basis of oxidative stress and inflammation in clinical settings, linking these pathways to the development and progression of cardiometabolic diseases. We also highlight the clinical relevance of antioxidant enzymes (e.g., SOD, GPx, CAT) and pro-inflammatory cytokines (e.g., IL-6, TNF-α, IL-1β) as measurable biomarkers that bridge experimental findings with patient outcomes.
We have also updated the clinical section to include a more detailed scientific explanation, highlighting the mechanisms, evidence, and importance of oxidative stress and inflammation in human diseases. Additionally, we focused the discussion on the main information we want to communicate, including the potential therapeutic uses of antioxidants and anti-inflammatory agents, supported by relevant clinical studies and trials.
-The author must show the major revisions made in the text by highlighting the changes in a different colored text. It is imperative to consider all these remarks to reinforce the manuscript's quality and conclude more accurately.
We thank the reviewer for this suggestion. All major revisions have been carefully incorporated into the manuscript, and the changes have been highlighted in colored text (red) to make them easier to identify. We have addressed all the remarks, including improving the clinical section, adding references, clarifying the limitations, and restructuring tables and figures to strengthen the manuscript’s scientific quality and lead to more precise conclusions.
Round 2
Reviewer 1 Report
Comments and Suggestions for Authors
The authors have addressed my concerns.
Author Response
We sincerely thank the reviewer for the positive feedback and for taking the time to review our revised manuscript. We're glad that our revisions addressed all the concerns.
Reviewer 4 Report
Comments and Suggestions for Authors
In this version of the reveiw, “Exploring the Interplay of Antioxidants, Inflammation, and Oxidative Stress: Mechanisms, Therapeutic Potential, and Clinical Implications” We can see an acceptable evolution compared to the first version because it has become more structured with more explanation.
the authors have taken the reviewer’s remarks and suggestions into consideration, which has positively impacted the quality and consistency of the review.
with this version, the review shows an good scientific level and represents an addedvalue in the interested research topics
the figures are still to be reviewed there are too basic and lacl innovation:
The author need to use a tool to properly prepare a figure suitable for the journal, for example, the elements in Figure 1 are not structured; figure 2, we cannot understand the principal information from this figure. Please revise accordingly.
Author Response
We sincerely thank the reviewer for the positive feedback on the revised version and for acknowledging improvements in structure, clarity, and scientific value. We also appreciate the constructive comments about the figures. In response, we have carefully revised the figures using professional design tools to make them more organized, scientifically accurate, and visually clear.
- Figure 1 has been simplified and restructured to emphasize key information, making it easier for readers to understand the central message quickly.
- Figure 2 has been redesigned to organize elements better and highlight main pathways and interactions with clearer visuals.
We believe these improvements significantly enhance the quality and innovation of the figures and better align them with the journal’s standards.